# Do People Care about the Origin of Wildlife? The Role of Social Stereotypes on Public Preference for Exotic Animals

**DOI:** 10.3390/ani12172160

**Published:** 2022-08-23

**Authors:** Rocío Alejandra Díaz, Verónica Sevillano, Marcelo Hernán Cassini

**Affiliations:** 1Laboratorio de Biología del Compoertamiento, Instituto de Biología y Medicina Experimental, Consejo Nacional de Investigaciones Científicas y Técnicas, Buenos Aires 1429, Argentina; 2Department of Methodology and Social Psychology, Facultad de Psicología, Campus de Cantoblanco, Universidad Autónoma de Madrid, 28049 Madrid, Spain

**Keywords:** social stereotypes, non-native species, animals

## Abstract

**Simple Summary:**

Many species have been introduced to regions outside their original habitat range. These non-native species are of great concern to conservation biologists, because they are considered to be dangerous to native species and ecosystems. However, the general public does not always agree with this appreciation and therefore conflicts are generated when trying to manage non-native species. This is one reason as to why it is important to understand the human dimension of this problem. We asked a group of college students about their favorite free-living animals and found that most preferred non-native species. To explain this result, we applied the theory of social animal stereotypes.

**Abstract:**

People’s attitudes to animals are becoming increasingly important for the success of invasive species management. We asked college students from Argentina to fill a questionnaire that included a question about their favorite free-living animal. A total of 159 responses were obtained. Native species were significantly less preferred than non-native species. We tested if these preferences were associated with animal stereotypes. The stereotype hypothesis predicts that animals from the contemptible stereotype (invertebrate, rodents, and reptiles) should be the least preferred taxa, and animals from the protective stereotype (pets, horses, and primates) should be the most preferred taxa; animals from the subordination (lagomorphs and birds) and threatening–awe stereotype (large carnivores) should show intermediate preferences. The first prediction was supported. However, students showed significant preference for non-native taxa included in the threatening–awe stereotype. We proposed that people prefer large carnivores (stereotypically strong, intelligent, and beautiful animals) when they are exotic, because they did not represent a risk.

## 1. Introduction

Many species have been introduced into regions outside their native habitat range. Many of these non-native species are harmless, but others can cause different degrees of damage to biodiversity and natural ecosystems [1]. Non-native species are defined as “invasive” when they threaten biodiversity, food security, health, or economic development [2]. For decades, invasive non-native species have been considered the second greatest threat to global biodiversity after habitat destruction [3]. They are also considered a major cause of economic losses worldwide [4]. Nevertheless, there is not a universal consensus among scientists around the native-non-native dichotomy. Davis et al. [5] published a seminal paper with a compelling title: ‘Don’t judge species on their origins’. The authors urged conservationists and land managers to focus much more on the functions of species rather than on where they are native or non-native. In recent years, the debate has intensified [3,6,7,8,9].

The social dimension of non-native species has been intensively investigated (reviewed by [10,11,12,13,14], among others). Attitudes to animals and their management are becoming increasingly important for the success of conservation and environmental initiatives [15]. One of the reasons is that conflicts frequently occur between different social actors. Several values, beliefs, or attitudes in relation to animals can determine different responses to the removal of non-native species that cause damage. While conservation biologists tend to agree that harmful species should be controlled, there are many non-specialists who do not consider these species to be a problem and are even opposed to the animals being eliminated [16,17,18,19,20,21,22,23,24].

In this study, we analyzed possible psycho-social factors involved in the preference of lay people about non-native species. A questionnaire was provided to students and recent graduates of psychology at the University of Buenos Aires, Argentina. This group can be considered as non-specialists regarding the invasive species issue. Among a variety of questions, we included: “which free-living animal do you like the most?”. We then separated preferred species according to whether they were native or exotic. This approach is different from the usual method in which the public is asked to select among an a priori list of species (e.g., [24,25,26,27,28]).

We tested the hypothesis of social animal stereotypes to explain preferences for animals. Some animal species are more positively perceived than others based on the characteristics socially attributed to them (i.e., social stereotype), which are not related to species origin but to the role of animals in society. Sevillano and Fiske [29] applied the Stereotype Content Model [30] to human-animal relationships. They defined four animal stereotypes: subordination (high warmth and low competence), threatening–awe (low warmth and high competence), contemptible (low warmth and low competence) and protective (high warmth and high competence) (Figure 1). They grouped animal species in accordance with these four categories: farm animals, lagomorphs, and birds are perceived similarly and were grouped in the subordination stereotype; large carnivores are included in the threatening–awe stereotype; invertebrate, rodents, and reptiles in the contemptible stereotype; and pets, horses, and primates, in the protective stereotype. Arguably, the beliefs associated with animal species can be ranked in relation to these four stereotypes [31]: the protective type is expected to include the most preferred animals because they elicited high ratings in both dimensions. The threatening–awe and subordination species should show an intermediate level of preference because of their ambivalent nature, with a positive perception in one dimension and a negative one in the other. The contemptible type should bring together the least preferred species because both dimensions show low ratings.

## 2. Methods

Data were collected at the Faculty of Psychology of the University of Buenos Aires by the first author (RAD). Each respondent was asked to participate voluntarily in the study, providing informed consent in order not to compromise the confidentiality and anonymity of their data (Table A1). The questionnaire included questions about demographic characteristics and environmental concerns that were analyzed in another publication [32]. This paper analyzes the answers obtained to the question about which free-living animal the respondents preferred. The responses to this open question were categorized according to two criteria: (i) whether the preferred species were native or exotic to Argentina and (ii) to which social stereotype the preferred species belonged.

Native species are those belonging to Argentinean fauna [33]. We used Darwin’s [34] definition of domestic animals, i.e., species that are the result of artificial selection processes. Domestic species were considered non-native or exotic species. When a response refers to a taxonomic category (e.g., felids) which includes species that could be both native and non-native, the response was categorized as ‘both’. Other methodological details can be found in Appendix A and Appendix B.

## 3. Results

A total of 159 responses of preferred species were obtained (Appendix C). Native species were significantly less preferred than non-native species, both when measured as frequencies of responses (χ^2^ (2) = 89.9, *p* < 0.00001) (Figure 2A) or number of preferred species (χ^2^ (2) = 10.9, *p* < 0.004) (Figure 2B). Among the first ten most preferred species (out of a total of 49 preferred species), seven were non-native (tiger, lion, horse, elephant, wolf, bear, and dog), three could be both native and non-native (deer, birds, and felids), and none were native. However, people who had the experience of living or those who already lived in the countryside showed a greater preference for native species than those who have always lived in the city (Figure 3, χ^2^ (2) = 7.1, *p* < 0.028).

There was a pattern of most preferred animals linked to the type of stereotype associated with the animal species. The most preferred species belonged to the threatening–awe stereotype and the least preferred animals were those belonging to the contemptible stereotype (χ^2^ (3) = 58.0, *p* < 0.00001) (see Figure 4). The species associated with the other two stereotypes showed intermediate preference. 

## 4. Discussion

Respondents expressed a strong preference for non-native species. This result agrees with those obtained in previous studies that found positive attitudes of non-experts towards non-native species. Boshoff et al. [35] found that most visitors to the Addo Elephant National Park in South Africa accepted the park having non-native species. Farnworth et al. [22] described substantial differences on the attitude towards the lethal control of eight non-native species in Australia: while conservationists routinely considered all species deserved control, the general public provided the lowest scores. Complementarily, the native nature of animals was not a main factor for non-experts. Fischer et al. [28], through a survey in eight sites across Europe, found strong relationships between beliefs about species and their control, in particular regarding their harmlessness and the desirability of an increase in this species. Other beliefs, such as perceived nativeness, were less influential. Moskwa [36] found that tourists did not show significant differences in their opinion regarding culling non-native and native species and only changed their opinion when given information regarding why animals may be culled. Nates et al. [37] found that young Argentinean rural student’s preferences and perceptions were strongly directed towards 18 non-native domestic species. Remmele and Lindemann-Matthies [38] found that German students perceived invasive alien species as beautiful and desired, especially mammals. In a study on student attitudes towards potential animal flagship species in Switzerland, Schlegel and Rupf [39] commented that, before the project started, most children showed preference for pets and exotic animal species. In North Carolina, USA, Schuttler et al. [40] found that children, whether they lived in urban or rural areas, preferred non-native mammals and were more likely to list local animals as scary than as liked. Ballouard et al. [41] found that French schoolchildren were more prone to protect exotic rather than local animal species.

The preference for animals varied significantly according to the four types of stereotypes. As predicted by Sevillano and Fiske [29], animals included in the contemptible stereotype were the species less frequently selected as the most preferred, while those included in the subordinate stereotype occupied an intermediate position. Instead, the expected result was not obtained for the other two types: the most preferred animals were those of the threatening–awe stereotype, while those of the protective stereotype showed intermediate frequencies. This mismatch in predictions may have been due in part to the way the question was asked, as only a preference for ‘free-living’ species was requested. Bearing in mind that several species included in the protective stereotype are domestic and can live in captivity, it could be possible that their lower representation was due to this bias in the question. Nevertheless, there were students who answered that their preference was for domestic animals such as dogs, cats, and horses, and these data were included in the analysis. Even with this methodological limitation, the remarkable preference for exotic wild species belonging to the threatening–awe stereotype requires an additional ad-hoc explanation.

Previous research indicated that people could have negative beliefs or emotions towards the animals included in the threatening–awe stereotype. For example, Jürgens and Hackett [42] proposed that negative feelings toward wolves are in part associated with aspects of wolf behavior, which corresponds to the human understanding of the notion of evil, due to a stereotype that may help fuel the heated societal debates about wolves. Sevillano et al. [43] identified spontaneous stereotypes of large carnivores in Spaniards that also included negative components. Therefore, we expected a medium or low preference for the animals included within the threatening–awe stereotype. However, species that do not inhabit the Neotropics such as tigers, lions, elephants, and wolves occupied the highest positions within the ranking of most preferred species of our interviewees. We propose an ad-hoc hypothesis for this unexpected preference, described in the following paragraph.

According to Sevillano and Fiske’s [29] model, each animal stereotype is defined by two socio-perceptive axes: warmth and competence (Figure 1). The threatening–awe stereotype is applied to species that are characterized by low warmth and high competence. On the one hand, they are perceived as unfriendly and even dangerous animals but, on the other hand, they are considered intelligent, strong, and beautiful (Sevillano et al. [43]). We propose that the low warmth component would be less accentuated in those countries where the species are exotic due to the fact that they would never represent a real danger for people as they do not inhabit the regions near them. Direct contact with these animals can only occur in zoos, where a distorted image of their dangerousness is fostered, since caged animals are perceived as tame and passive compared to animals in the wild [44]. In contrast, these non-native animals are salient in Western societies, since they are frequently used as symbols of strength, agility, or intelligence [37]. Based on this hypothesis, we propose that the preference for non-native animals belonging to the threatening–awe stereotypes expressed by the Argentine public is due to the fact that, in places where these species do not naturally inhabit, values of warmth and competence increase in relation to those places where these species are native. In other words, our findings suggest that the exotic origin of a species could favor the development of species preferences by overshadowing their negative traits.

## 5. Conclusions

The results of this study may have relevance in the management of animal diversity. Even when the importance of the human dimension in this management has long been recognized, invasion biology is still dominated by a ‘top-down’ approach in which ‘experts’ define the problem, evaluate the evidence and management options, and advise decision makers, who must then persuade ‘the public’ to accept their decisions, justifications, and supporting evidence [19,45]. It is difficult to imagine that this top-down approach would work in a context in which lay people have strong preferences for non-native species or express favorable emotions towards species that should be eliminated, because they would be causing harm. This difficulty will be even greater if those preferences, opinions, emotions, or attitudes towards non-native species are based on deep psychological mechanisms such as moral principles or social stereotypes. Furthermore, if the opinion of the non-experts coincided in downplaying the geographical origin of the species, the experts should reanalyze the validity of the axiom that non-native species are all potentially harmful. The message given by the public in this study and others appears to be that conservationists should be worried about the negative impacts of species independent of their native status.

## Figures and Tables

**Figure 1 animals-12-02160-f001:**
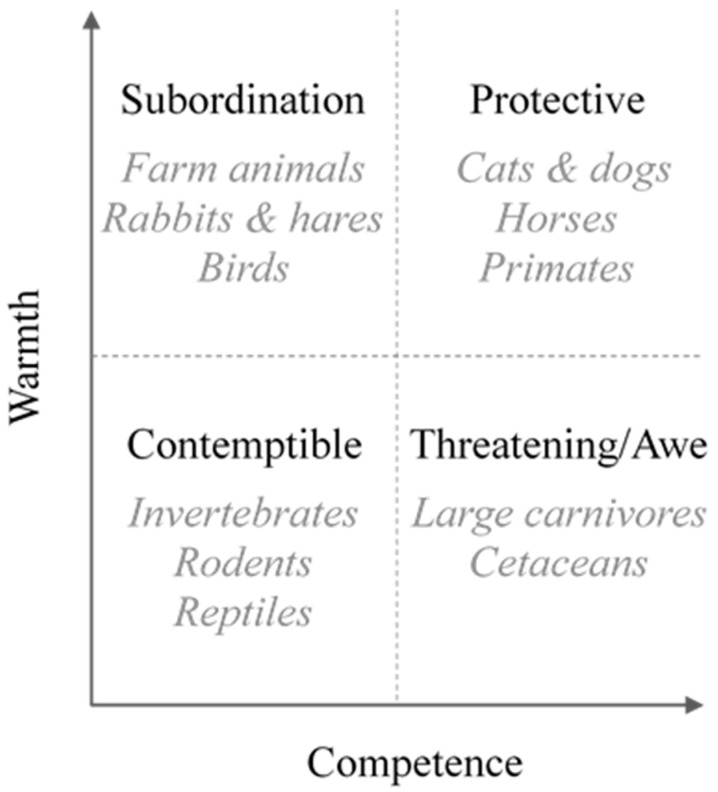
Sevillano and Fiske’s [29] model of animal stereotypes. Description in the Introduction.

**Figure 2 animals-12-02160-f002:**
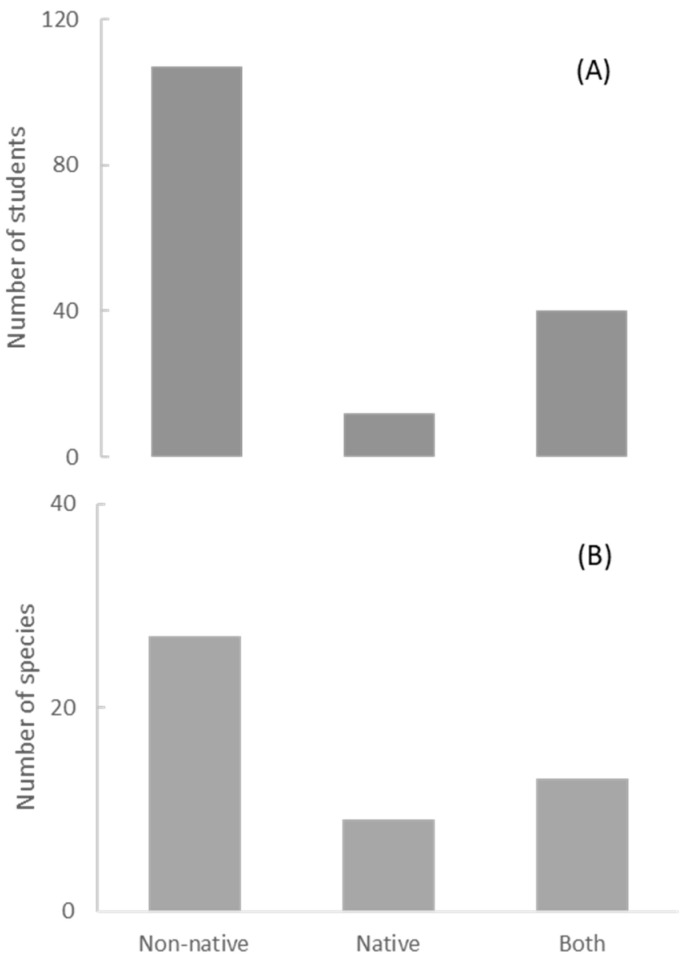
(**A**) Frequencies of responses and (**B**) number of different taxa, when the preferred species was a non-native taxon, a native taxon, or a taxon that could be regarded as both native and non-native species.

**Figure 3 animals-12-02160-f003:**
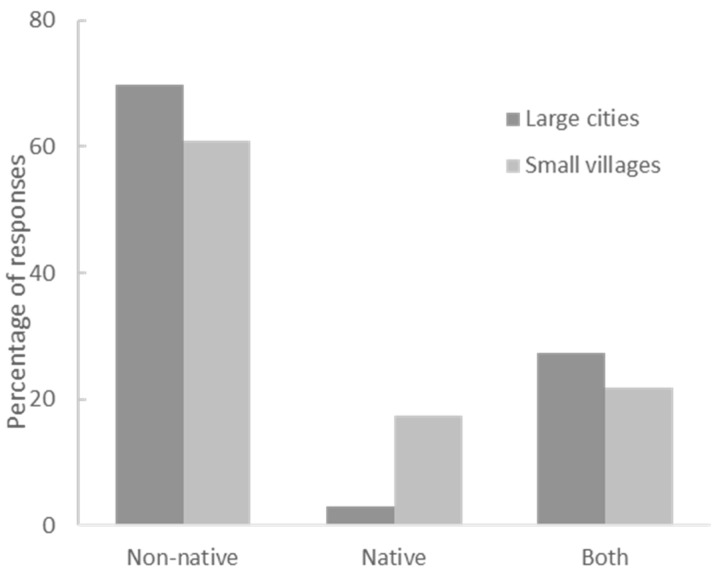
Percentages of responses for which the preferred species were native, non-native, or taxa that included both categories, comparing respondents who had always lived in large cities with those who had once lived or lived in small villages. Those who once lived in the countryside showed a significantly greater preference for native species.

**Figure 4 animals-12-02160-f004:**
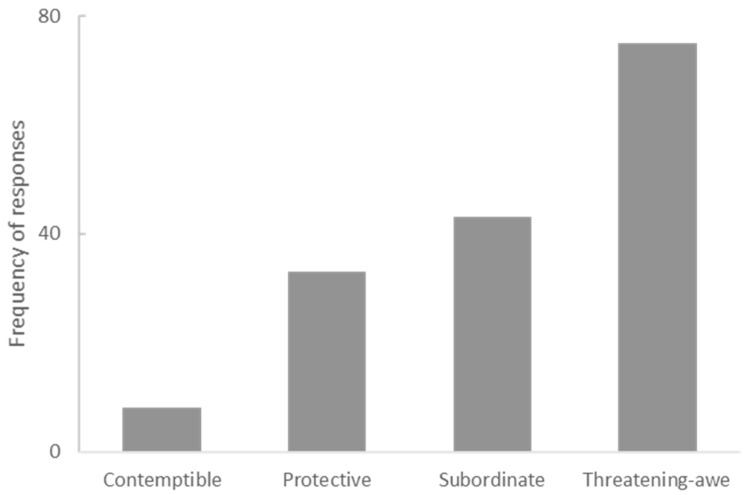
Percentages of responses for which the preferred species could be assigned to the four animal stereotypes defined by Sevillano and Fiske [29].

## Data Availability

Data provided in the Appendices.

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
