# Peer review of "Do People Care about the Origin of Wildlife? The Role of Social Stereotypes on Public Preference for Exotic Animals"

_animals, 2022, doi:10.3390/ani12172160_

Round 1
Reviewer 1 Report
The article the authors are presenting is well-structured, the topic interesting, and the results are clearly reported and discussed.
However, I found some issues that need to be addressed:
Materials
- Although the survey is analyzed in another publication (Diaz et al., 2020), I think the specific question needs to be reported.
- in general, some details about the survey and how the data have been handled and analyzed.
-The citation "Diaz et al., 2020" is not reported in the reference list.
Results
- I would like more information about the respondents and their answers. Were they all native of Argentina? Age?
- What is the range that defines "large cities" and "small villages"? And "countryside" is the same as "small village"?
- Figures 3 and 4 need to be rearranged in order.
Author Response
Although the survey is analyzed in another publication (Diaz etal., 2020), I think the specific question needs to be reported.
Response: We included the whole questionnaire as a Supplementary material 2
- in general, some details about the survey and how the data have been handled and analyzed.
Response: We added an Appendix with additional information on methodology
-The citation "Diaz et al., 2020" is not reported in the reference list.
Response: Done
Results
- I would like more information about the respondents and their answers. Were they all native of Argentina? Age?
Response: Added as an Appendix
- What is the range that defines "large cities" and "small villages"? And "countryside" is the same as "small village"?
Response: Added in the Appendix
- Figures 3 and 4 need to be rearranged in order.
Response: sorry but we did not find this problem
Reviewer 2 Report
In this article, the authors present and discuss the results of a survey on people’s favourite undomesticated/free-living animals with particular interest in their attitudes towards non-native species.
The authors designed a questionnaire in which they asked 159 Argentinian psychology students and graduates – among other things – to name their favourite free-living animal. The reported species was classified according to the “Stereotype Content Model” by Sevillano and Fiske which defines four stereotypes of human-animal relationships. The authors argue that the degree of preference for certain animals correlates with the classification in the model, i.e. the most preferred animals can be classified as “protective”; on the intermediate level of preference, there are those animals with a relationship classified as “subordination” or “threatening/awe”; and the relationship to the least preferred is defined as “contemptible”. Furthermore, the reported species was classified as “native” (to Argentina), “non-native” or “both” (if a group of animals was named that comprises both native and non-native species). The results show that most study participants named species that are non-native to Argentina and belong to the “threatening/awe” relationship type of the model.
The authors discuss that, in accordance with findings from the literature, most members of the public have a strong preference for non-native species. Additionally, the authors were surprised to find that, against the prediction of the above-mentioned model, most preferred species belonged to the threatening/awe stereotype rather than the protective stereotype. It is argued that certain properties of the model (here: the “low warmth component”) do not apply in the study because the mentioned species are not native to Argentina and, e.g., the threat posed by large carnivores like lions or tigers is not perceived as relevant. The authors come to the final conclusion that the positive public attitude to non-native species may present a relevant factor in ecological management when discussing measures involving so called invasive species.
While it is addressing an issue of high importance – public attitudes towards invasive species – there are several points that should be clarified before publishing the results of this study.
l.8: In the abstract, you claim that you interviewed students. Later you write about a questionnaire. Please clarify.
l. 48: Why did you ask psychology students? They are certainly not the only group that can be considered non-specialist regarding your questions.
l. 50: What kind of questions were included in the questionnaire? This might be important, as they might have led participants (not) to think of certain species. You are referring to another publication (l. 81) but it would be helpful to summarise the content of the other questions, here.
l. 50: The expression “wild animal” is connoted in a non-neutral way, which might have motivated your participants to name species like lions or tigers rather than butterflies or mice. A more neutral wording would have been “free-living” animal. You should discuss the (potential) effect of your choice of words in your article.
l. 67: What do you mean by “positive evaluation or preference”, here? How are animals evaluated and what kind of preference do you refer to in your study? This is important for your further argumentation.
l. 98 f. : Among your top-ten list, there are already three groups of animals that do exist as free-living animals but are most likely named as companion animals, here (horse, cat, dog).
l. 114: You should rephrase that, because you only asked for naming one species! "Least preferred" sounds as if there had been a ranking with preference values for every group of animals. Actually, those participants naming, e.g., “lion” as their most preferred animal, did not tell you to what extent they favour snakes or spiders.
l. 130: This passage is not clear to me: What kind of relationships between which beliefs were found and how does that affect your point regarding nativeness? Please phrase more clearly!
l. 142: Your survey excluded domestic animals, so this aspect is not comparable
l.148: Again: your participants could only name one species, so it is hard to tell which they preferred least.
l. 149: Submissive = subordinate ?
l.152-160: To me, this explanation sounds quite straightforward: You did not ask for species that fall into the "protective" category, so those were not (or hardly, and if, then by mistake!) mentioned. You should clearly address that the model includes groups of species that you explicitly tried to exclude in your study, which is why the classification according to the models has strong limitations for your discussion.
l. 170: Again: It was not a ranking of species. The participants merely reported their one favourite species and did not tell you their preferences regarding any other animals.
l. 173: What do you mean by “non-native animals are part of the cultural heritage of the region”?
l.180: Is there any support in the literature that in regions with native large carnivores the attitudes are different? Are, e.g. the attitudes towards lions in regions with native lions similar to the ambivalent attitudes towards wolves in regions with native wolves?
Some more general comments:
You report that your study is exceptional because it does not provide a list of species that are to be ranked by the participants but you directly ask for their one favourite species of free-living animal (l. 51). This approach, however, does present the problem that you cannot tell anything else about the participants’ preferences – neither regarding their reasons nor regarding their attitudes towards other species. An interesting question would be if your data are, indeed, specific for Argentina or if, and that is my hypothesis, the species on your top-ten list would be named by participants from quite different countries. Lions and tigers are ubiquitous, not only in documentaries and post cards, but in children’s books and toys, names and logos of sports clubs, movies etc. If that is considered a reason for or a consequence of many people’s favouritism remains to be discussed. Furthermore, the attitude towards invasive species – and that might be an opportunity to refer to the stereotype content model again – is certainly framed by the question what kind of species is dealt with. Jumping from naming one favourite free-living species to the participants’ potentially complex attitude towards invasive species should be explained in more detail.
Nevertheless, your research question is really important and I would really like to read a publication addressing this topic!
Author Response
l.8: In the abstract, you claim that you interviewed students. Later you write about a
questionnaire. Please clarify.
Response: In the Abstract, we changed to: We asked college students from Argentina to fill a questionnaire that included a question about their favorite free-living animal.
- 48: Why did you ask psychology students? They are certainly not the only group that can be considered non-specialist regarding your questions.
Response: It was a convenience sample
- 50: What kind of questions were included in the questionnaire? This might be important, as they might have led participants (not)to think of certain species. You are referring to another publication (l. 81) but it would be helpful to summarise thecontent of the other questions, here.
Response: Added as an Appendix
- 50: The expression “wild animal” is connoted in a non-neutral way, which might have motivated your participants to namespecies like lions or tigers rather than butterflies or mice. A more neutral wording would have been “free-living” animal. You should discuss the (potential) effect of your choice of words in your article.
Response: Thank you for providing us with a solution to a translation problem. In the actual questionnaire, we asked for the favorite 'silvestre' animal. This word in Spanish has not a precise English translation, thus we had used 'wild'. However, 'wild' in Spanish is translated as 'salvaje'. In Spanish you can say 'planta silvestre' but you cannot say 'planta salvaje', because 'salvaje' refers only to animals. 'Free living' is the best translation to 'silvestre'.
- 67: What do you mean by “positive evaluation or preference”,here? How are animals evaluated and what kind of preferencedo you refer to in your study? This is important for your furtherargumentation.
Response: we changed the sentence to: ' the beliefs associated to animal species can be ranked in relation to these four stereotypes (Sevillano & Fiske 2019)'
- 98 f. : Among your top-ten list, there are already three groups of animals that do exist as free-living animals but are most likely named as companion animals, here (horse, cat, dog).
Response: We am not sure if we understand your comment. Many domestic mammals are actually some of the most dangerous invasive species and many 'wild' animals are used as pets or breeding in order to exploit their meat, fur, feathers, etc. Thus, there apparent distinction is not always so clear, especially when the analysis is oriented towards the comparison between invasive and non-invasive animals.
- 114: You should rephrase that, because you only asked for naming one species! "Least preferred" sounds as if there had been a ranking with preference values for every group of animals. Actually, those participants naming, e.g., “lion” as their most preferred animal, did not tell you to what extent they favour snakes or spiders.
Response: changed to 'most preferred animals'
- 130: This passage is not clear to me: What kind of relationships between which beliefs were found and how does that affect your point regarding nativeness? Please phrase more clearly!
Response: we are sorry but there was a part of the sentence missing. The complete sentence is: 'Fischer et al. (2011), through a survey in eight sites across Europe, found strong relationships between beliefs about species, in particular regarding their harmlessness, and the desirability of an increase in this species'.
- 142: Your survey excluded domestic animals, so this aspect is not comparable
Response: I think that perhaps you are confused about the concept of 'domestic' animal. Domestication is not related to the place where the animal live but is related to its genotypic and phenotypic traits. An animal is domestic when some of these traits evolved by artificial selection (i.e., by human-induced selection). A free-living domestic animal is normally called 'feral' animal. Pigs and horses are considered domestic animals even when they live in the wild and reproduce freely. A wild animal has all its traits evolved by natural or sexual selection.
l.148: Again: your participants could only name one species, so itis hard to tell which they preferred least.
Response: changed to 'were the species less frequently selected as the most preferred'
- 149: Submissive = subordinate?
Response: change submissive to subordinate
l.152-160: To me, this explanation sounds quite straightforward: You did not ask for species that fall into the "protective" category, so those were not (or hardly, and if, then by mistake!) mentioned.You should clearly address that the model includes groups of species that you explicitly tried to exclude in your study, which is why the classification according to the models has strong limitations for your discussion.
Response: Again, we are not sure what is the meaning of your comment. We added a few changes trying to improve clarity. However the main point were already mentioned. Protective category includes include wild animals together with domestic ones. However, these wild species were not selected significantly by respondents, and that deserves an explanation
- 170: Again: It was not a ranking of species. The participants merely reported their one favourite species and did not tell you their preferences regarding any other animals.
Response: we agree with you that we sometimes misused the work 'preferences' but we disagree with your comment that it is not a ranking of species.
- 173: What do you mean by “non-native animals are part of the cultural heritage of the region”?
Response: We removed the sentence because the explanation of the hypothesis is in the following paragraph
l.180: Is there any support in the literature that in regions with native large carnivores the attitudes are different? Are, e.g. the attitudes towards lions in regions with native lions similar to the ambivalent attitudes towards wolves in regions with native wolves?
Response: No, we did not find such literature
Some more general comments:

You report that your study is exceptional because it does notprovide a list of species that are to be ranked by the participantsbut you directly ask for their one favourite species of free-livinganimal (l. 51). This approach, however, does present theproblem that you cannot tell anything else about the participants’preferences – neither regarding their reasons nor regarding theirattitudes towards other species. An interesting question would beif your data are, indeed, specific for Argentina or if, and that ismy hypothesis, the species on your top-ten list would be namedby participants from quite different countries. Lions and tigers areubiquitous, not only in documentaries and post cards, but inchildren’s books and toys, names and logos of sports clubs,movies etc. If that is considered a reason for or a consequenceof many people’s favouritism remains to be discussed.Furthermore, the attitude towards invasive species – and thatmight be an opportunity to refer to the stereotype content model again – is certainly framed by the question what kind of speciesis dealt with. Jumping from naming one favourite free-livingspecies to the participants’ potentially complex attitude towardsinvasive species should be explained in more detail.
Response: In the presentation of the hypothesis, we eliminated all reference to Argentina since, like you suggests, this phenomenon, if it existed, would not be limited to this country
Reviewer 3 Report
Very interesting angle to take for this paper as a way to use data already collected and analyzed to answer different questions. There are some typos and awkward language in a few places. In the abstract, line 7 should read "People's attitudes..." and in line 10, it should say "associated with animal..." In line 82, you need to remove the duplication (the result obtained). In line 116, it should say "associated with animal..."
I appreciate that the authors did not unnecessarily belabor points or add in redundant text to fill space, This was a nice succinct summary of their project and findings.
Author Response
Very interesting angle to take for this paper as a way to use data already collected and analyzedto answer different questions. There are some typos and awkward language in a few places. Inthe abstract, line 7 should read "People's attitudes..." and in line 10, it should say "associatedwith animal..." In line 82, you need to remove the duplication (the result obtained). In line 116, itshould say "associated with animal..."
Response: all these typos were corrected. Thank you for your kind comments